# Trecode: A FAIR Eco-System for the Analysis and Archiving of Omics Data in a Combined Diagnostic and Research Setting

Hindrik HD Kerstens [†], Jayne Y Hehir-Kwa [†], Ellen van de Geer, Chris van Run, Shashi Badloe, Alex Janse, John Baker-Hernandez, Sam de Vos, Douwe van der Leest, Eugène TP Verwiel, Bastiaan BJ Tops and Patrick Kemmeren *

Princess Máxima Center for Pediatric Oncology, 3584 CS Utrecht, The Netherlands
* Correspondence: p.kemmeren@prinsesmaximacentrum.nl; Tel.: +31-88-972-7272
† These authors contributed equally to this work.

**Abstract:** The increase in speed, reliability, and cost-effectiveness of high-throughput sequencing has led to the widespread clinical application of genome (WGS), exome (WXS), and transcriptome analysis. WXS and RNA sequencing is now being implemented as the standard of care for patients and for patients included in clinical studies. To keep track of sample relationships and analyses, a platform is needed that can unify metadata for diverse sequencing strategies with sample metadata whilst supporting automated and reproducible analyses, in essence ensuring that analyses are conducted consistently and data are Findable, Accessible, Interoperable, and Reusable (FAIR).We present "Trecode", a framework that records both clinical and research sample (meta) data and manages computational genome analysis workflows executed for both settings, thereby achieving tight integration between analysis results and sample metadata. With complete, consistent, and FAIR (meta) data management in a single platform, stacked bioinformatic analyses are performed automatically and tracked by the database, ensuring data provenance, reproducibility, and reusability, which is key in worldwide collaborative translational research. The Trecode data model, codebooks, NGS workflows, and client programs are publicly available. In addition, the complete software stack is coded in an Ansible playbook to facilitate automated deployment and adoption of Trecode by other users.

**Keywords:** next-generation sequencing; FAIR; data management platform; data analyses; data provenance; automation

## 1. Introduction

The analysis of next-generation sequencing (NGS) data poses significant challenges for bioinformaticians. First, as NGS technologies continue to become cheaper, faster, and more reliable, they are increasingly used both in research and diagnostics [1,2]. Current diagnostic utilization of NGS includes a wide range of applications, ranging from DNA variant discovery to gene fusion detection by transcriptome assessment (RNA-Seq) and DNA methylation profiling. In addition, NGS technologies are also used within cancer research, for detailed genetic characterization of patient samples to study the molecular mechanisms that drive malignancies and to develop personalized treatment [3–5]. These developments are only possible if bioinformatic and data services can accurately and efficiently process, store, and distribute highly diverse NGS data [6]. Automated data handling is required to match the increasing scale in which data are generated and to effectively use the expanding variety of compute and data storage infrastructures. Furthermore, automation is essential to enable a relatively small number of bioinformaticians to process the data with minimal chance of human errors whilst using compute resources efficiently. If automated sufficiently, bioinformaticians can focus on method development and result interpretation rather than performing the analyses, also with increasing workloads.

Secondly, (meta) data management is typically spread across separate clinical and/or research database(s), each using different data models. These systems are often loosely coupled to genome analysis platforms, which tend to be monolithic workflows designed to support either clinical or research analysis. Scalable genomics analysis requires portable analysis workflows that can be implemented across a variety of software and hardware environments, including high performance computing clusters and cloud instances. Domain-specific workflow languages [7,8] and executers [9,10] are being increasingly used to simplify analyses across a variety of execution environments. This progress facilitates a paradigm shift of moving analysis workflows to the data rather than the data to the analysis workflow. The latter is becoming increasingly difficult given the increasing volume of data. Furthermore, it fuels the development of standards for describing computational workflows with the aim of making genome analysis reproducible and transferrable to other labs [11].

The research of rare diseases and the development of precision medicine requires combining data between institutes for large-scale analyses. Useful and sufficient metadata must be provided to enable findability, reusability, and correct interpretation of the distributed data. However, genomics is a dynamic field, and capturing metadata for emerging analysis techniques is a moving target, placing demands on the flexibility of the data models used in the genomics platform. Furthermore, the genomics platform should be accessible and interoperable to allow programmatic exchange of metadata, thereby avoiding human error and increasing scalability.

The *Princess Máxima Center for Pediatric Oncology* is the Dutch national children's oncology center and started sequencing research samples in 2016 and sequencing based diagnostics in 2018. For analyzing and tracking all sequencing data produced in the center, we have developed a platform called Trecode, which is presented here. As of November 2022, this platform has archived the metadata and primary analysis of 3000 sequenced transcriptomes and more than 2600 whole exome and genome sequenced samples. Subsequent somatic variant analyses have been completed on 1295 tumor-normal sample pairs for clinical (WXS) and 1188 for research (WGS). With approximately 600 children per year being affected by cancer in the Netherlands, this data collection will potentially grow by this number of sample pairs yearly. Trecode is a generic data management and analysis platform that allows a small team of 4 to 6 bioinformaticians to support all sequencing analysis related activities in both routine clinical diagnostics and research biobanking. Trecode integrates sequencing experiment description and computational analyses into a single data model, facilitating sample relation tracking and automated genome analyses. Furthermore, this integration promotes data exchange between users from different disciplines. The Molgenis scientific data platform [12] is used to make this data model interoperable, which aids in removing barriers in Findability, Accessibility, Interoperability, and Reusability (FAIR) of genomics data.

The metadata standards enforced by the Trecode platform facilitates automated submissions to the European Genome-phenome Archive (EGA) [13] public repository. The computational analyses are fully integrated and designed with the aim of maximizing code re-use, reproducibility, and transferability. The Trecode platform assists operators and bioinformaticians in performing reproducible data analyses, and the generic, flexible, and scalable design allows developers to adjust the platform with little effort.

## 2. Materials and Methods

### 2.1. Softwares

Trecode is implemented using Molgenis, an open source scientific data management platform [12], and Cromwell Workflow Description Language (WDL) executer [10]. This creates a platform with a flexible and extensible data model that is tightly connected to an analysis workflow execution service using a REpresentational State Transfer (REST) protocol [14]. The database is interoperable through the Molgenis web-based graphical

user interface (web GUI) and its REST based application programming interface (API), which offers accessibility to a broad range of end users.

### 2.2. Datamodel

We based our data model on SRA/EGA [13,15] and extended with attributes from the Investigation/Study/Assay (ISA) open source metadata tracking framework [16]. For describing the automated computational workflows, we created additional entities and attributes.

### 2.3. Identifiers

Trecode identifiers have a human recognizable pattern and show similarity to the accessions used in the Encode project [17]. The identifiers are in a format where the first three letters of the identifier define the institute and origin of the data source, followed by a two-letter code that represents the metadata type (Table S1). Trecode identifiers end with a six-character namespace, of which the first three are digits and the last three are letters, resulting in more than 17 million unique identifiers per data source or entity type, for example: PMCRX001ABC has the Princess Máxima Center (PMC) as institute of origin and is of metadata type Experiment (Table S1).

### 2.4. Workflow Language

Trecode uses workflow description language (WDL) [7] to define workflows that are executed by the Cromwell executer [10]. The main considerations for using this workflow description and execution software are human readability of the domain-specific code, compute backend agnosticism, task result caching, error reporting, and the availability of a REST API. The advantages of using a workflow executer are that explicit calls to the compute backend can be omitted, keeping the code base small and maintainable while maintaining the portability of the workflows from, e.g., "classical" on-premises high-performance compute to cloud-based compute environments. The Cromwell workflow execution service will translate the implicit request for compute resources to specific calls on the underlying compute infrastructure.

## 3. Results

We designed Trecode to enable the systematic collection, annotation, and genomic characterization of samples for both tumor and healthy tissue of all patients entering a research hospital. RNA sequencing and WXS was performed systematically on all diagnostic samples. Once informed consent was given, these samples additionally became part of the biobank and WGS was performed.

### 3.1. Platform Design

Trecode captures all required information; models relations between samples, their derivatives, and NGS results in a single data model; and performs genomics analyses semi-automatically (Figure 1).

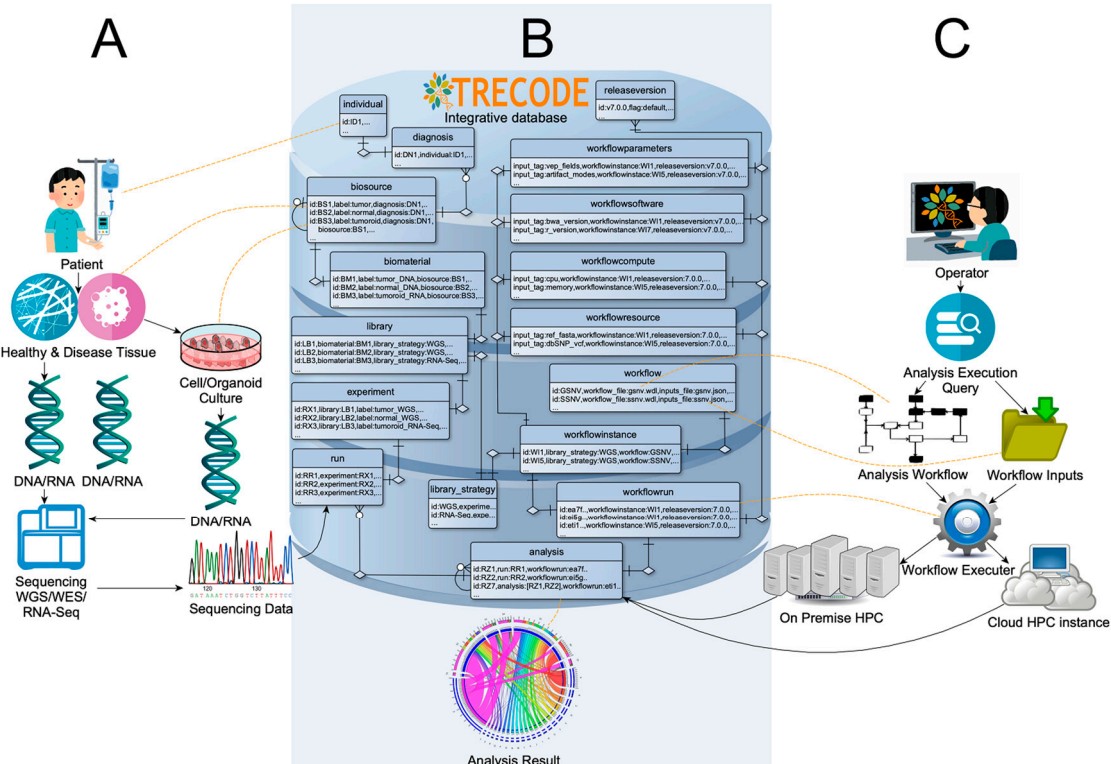

**Figure 1.** Overview of the Trecode platform. (**A**) Depicts a graphical representation of the biobanking activities. (**B**) The Trecode data model, of which the core entities are shown, describes the wet-lab experiment as well as the computational setup in detail and interconnects the two. Trecode makes experimental data programmatically accessible to operators, records the computational analyses performed, and takes care that analyses results are consistently linked back to the experimental data. For clarification, some tables are linked by dotted orange lines to the biobank or computational entity that they represent. (**C**) Depicts the architecture for computational analyses.

Metadata and NGS analyses results remain findable and are kept accessible via the platform's standardized interfaces. NGS result files are stored using community-developed standard formats [18] in Trecode and are annotated using well-established commonly used ontologies [19], ensuring optimal reusability according to the FAIR principles [20]. Based on pre-existing open software components, we aim for the maximum reuse of current developments.

Samples collected from an individual are identified as (primary) biosources from which new biosources can be derived by means of cell or organoid culturing. Extracts isolated from a biosources such as DNA or RNA are termed biomaterials. These are stored as a separate entity and can be traced back to the originating biosource and individual using the data model. Within Trecode, biomaterials are the primary inputs for generating sequencing libraries. A sequencing experiment, such as whole genome (WGS), whole exome (WXS), or transcriptome (RNA-Seq), requires a context-specific sequencing library and results in a sequencing run with associated sequencing run data (Figure 1, panel A).

An operator initiates the data analyses by querying the Trecode instance for the existence of sequencing data based on (i) the sequencing strategy (WGS/WXS/RNA-Seq), (ii) for a (set of) biomaterial(s), and (iii) the type of analysis to be performed (Figure 1, panel C). As part of the query results, a predefined bioinformatics workflow for genome sequencing analysis is initiated using a set of sequencing strategy-specific parameters. Parameters include required genomics and compute resources as well as workflow computational parameters that together form a complete list of workflow inputs. The workflow is then executed using an execution service, which translates generic workflow task defi-

nitions into specific compute backend calls. By using an executer that supports multiple compute backend services, analyses workflows become transferrable and can be executed on a variety of compute infrastructures. Workflows not only include tasks for data analysis but also tasks for registering and linking performed analyses to biomaterial(s) in the database and archiving analysis results in a file or object store.

## 3.2. Data Model

In Trecode, samples and their relationships are recorded concisely, together with the NGS sequencing experiments performed (Figure 2).

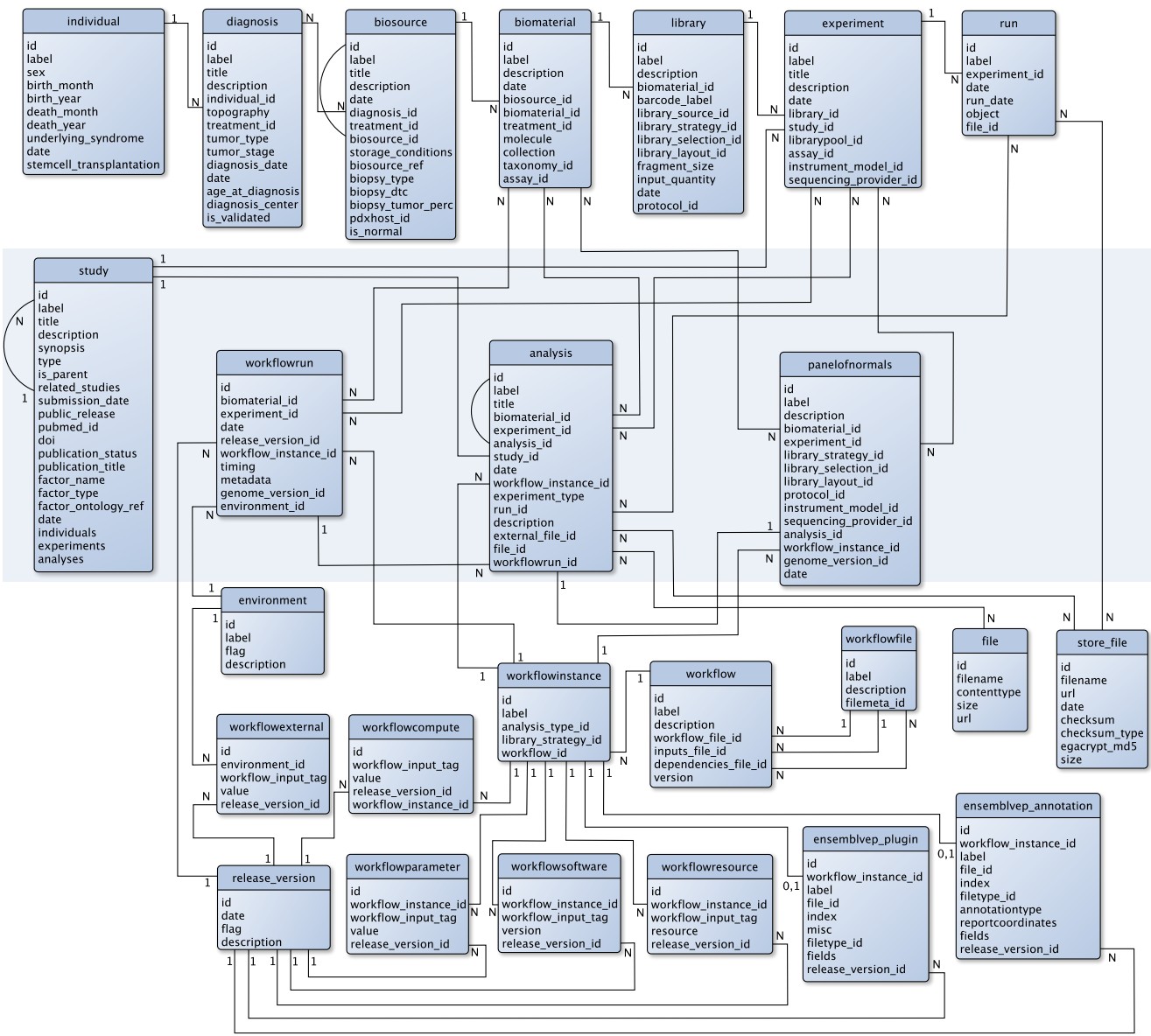

**Figure 2.** Overview of the core database model without vocabulary/ontology codebooks and Molgenis system tables. (top section) Tables describe the samples and their relationships in the wet-lab sequencing experiments; (bottom section) Tables describe the computational analysis workflows including the genomics resources (references, variant databases) and computational resources required for workflow execution; The shaded tables (middle section) are data integrating tables that describe sets of wet-lab experiment data or the results of computational analysis.

Moreover, the genomics analyses created through workflows are interlinked with the primary 'raw' data and analysis result files. The core sequencing data tables are: 'study', 'individual', 'diagnosis', 'biosource', 'biomaterial', 'library', 'experiment', 'run', and 'analysis'. A 'study' can be nested, using 'sub-studies', and is related to 'individual' by the 'individual study' cross table (Figure S1). The data model ensures that both experiments and computational analyses are linked to studies.

An 'individual' can have one or more malignancies that are each recorded in Trecode as 'diagnosis' and is referenced from 'biosources' that represent samples from individuals, such as tumor and normal tissue/blood samples. An essential task of the 'diagnosis' entity is to not only describe the clinical diagnosis but also to link related biosources, such as tumor and normal samples, relapse tumor samples, or refractory cancer events from a single malignancy. Biomaterial is the biosource extract that is analyzed in NGS and is referenced from 'library' or 'assay', representing the sequencing library that is created from the extracted DNA/RNA or the assay that is performed, e.g., for methylation analysis (Figure S2). The wet-lab sequencing or array-based analyses strategies in 'library'/'assay' is further referenced from 'experiment', which holds details of the experimental setup. Entity 'experiment' is referenced from 'run', which holds links to the raw sequencing/methylation data, and from 'analysis', which describes the computational processing of the sequencing/methylation results using 'library-' or 'assay-'specific workflow configurations.

In addition, Trecode stores analyses workflows, their requirements, and dependencies as well as analyses results created by each workflow execution. The series of computational processing steps performed by a specific bioinformatic workflow is described in the table 'workflow'. Attribute values for 'analysis_type_id' and 'library_strategy_id' in table 'workflowinstance' determine for which experimental setups a generic bioinformatic workflow is used. The initiation of a generic bioinformatic workflow is configured in supporting workflow tables, which refer to 'workflowinstance', namely 'workflowsoftware', 'workflowparameter', 'workflowresource', and 'workflowcompute'. These tables select software (versions), the workflow and task level parameter values, genomics resources (references, genome annotations, variant database versions) to be used, and the compute resources to be requested. As new versions of workflows are released, updates such as parameter changes remain traceable by tagging these new settings with a release version that is defined in the table 'release_version'. This concept is implemented across all supporting workflow tables. Additionally, the specific code for a workflow is described in the table 'workflowfile'. Compute environment differences are captured by parameters in the Trecode tables 'environment' and 'workflowexternal'. To promote consistent and uniform naming of workflow and task level inputs and support reuse, all input tags used in workflows and tasks are described in Trecode.

The computational analyses for (a set of) sequencing experiment(s) using a 'workflowinstance' is logged in the table 'workflowrun'. This table links the workflow execution products (analysis results), which are described in the linked table 'analysis'. All resulting files from a workflow run are registered in the table 'data_file_store'. Computational analyses can be performed on biomaterials' primary data via the linked experiments but can also be a secondary analysis that uses primary analyses results as input. In our primary analysis, the sequencing reads of an experiment are mapped to the reference genome, and germline variants are detected. In a secondary analysis, a pair of primary analysis mapping results are used as input and compared as in, for example, somatic single nucleotide variant analysis.

Within Trecode, panels of normals are created using dedicated workflows and stored in the table 'panelofnormals' with a reference to the generic workflow analysis results table 'analysis'. As a result, exactly how, and which, samples are included in the panels of normals remains traceable. Panels of normals can subsequently be queried to be used for artifact reduction [21] purposes in, e.g., copy number variant (CNV) and somatic single nucleotide variant (SNV) analyses.

Once the analysis is completed for a series of experiments, the data can be submitted to a sequencing repository to make it publicly accessible. The Trecode platform provides tools to perform automatic submissions to EGA [13] by collecting raw sequencing data, analysis results, and required metadata, given a number of experiment IDs (Figure S3a). The data model enforces researchers to collect sample metadata from the beginning of the project, is designed to perform submissions in a highly automated fashion, and tracks what has been submitted (Figure S3b).

### 3.3. Accessibility to Users from Multiple Disciplines

The Trecode platform links experimental and computational metadata as well as workflow execution code in a single data model. Consequently, the platform is useful to a broad range of users, in particular when users require exchange of information across research fields. Data analysis operators are provided with sufficient experiment (meta) data to perform computational analyses. Wet-lab scientists will find the computational analyses results linked to their experiments. Developers can exploit the data model flexibility and programmatic interfacing to efficiently add or improve functionality. By providing both a web GUI and REST API, the Trecode platform keeps data and analyses accessible for all users in a broad variety of roles.

### 3.3.1. Operators

The platform provides genomics data, metadata, data analyses pipelines (workflows), and workflow management functionality to operators who routinely perform genome analyses. Operators typically insert metadata for new samples and subsequently execute workflows to analyze these samples. Newly inserted sample metadata is checked for consistency and vocabulary by the Trecode platform and is implicitly linked to default analysis workflow instances.

Computational analysis queries are checked for sensibility by the platform command line client to avoid obvious human errors. For example, during somatic variant analysis, it is checked if the specified biomaterials originate from a single individual. In general, tumor versus normal comparisons are made in somatic analyses. Deviating queries such as tumor versus (relapse) tumor result in a warning where the operator may explicitly indicate that this check should be ignored. Furthermore, queries are checked as to whether library strategy and sequencing platform correspond between the queried sequencing experiments and the panel of normals being used. Once the input is accepted, the workflow will be started and the platform provides the operator with information about the workflow progress and consistent error reporting within a few mouse clicks, and it ensures that created analysis products become findable and linked to the sample metadata. Linked analysis products such as quality control (QC) metrics can be directly viewed, whereas other products can become the subject of secondary analyses or can be used in reference panels, facilitating the analysis plans of the operator.

### 3.3.2. Wet-Lab Scientists

Trecode's web GUI with multiple search functions and wet-lab intuitive data model allows lab technicians to easily find information about samples registered and processed, as well as associated metrics. Sample level QC metrics are a key resource for lab technicians to review and benchmark laboratory protocols. Furthermore, the platform offers browse and lookup functionality from individual to experiment level, and the progress of computational analyses can be followed in real-time. More complex sample relations can also be traced via the web GUI, and Molgenis' plugin architecture supports the creation of additional views on the data as desired.

### 3.3.3. Bioinformaticians

Bioinformaticians and developers interact with the Trecode platform when updating existing or implementing new workflows and registering genomics resources. This user

group typically uses the REST API of the platform to make workflow and resource related changes programmatically. The REST API is also used by bioinformaticians who query experiment and analyses results tables to perform aggregate analyses based on a custom subset of primary analyses results.

### 3.3.4. Data Security

Data is protected in Trecode using a password-based login, and role-based security is used to limit access or restrict operations to authorized users (Table 1).

**Table 1.** Roles and their permissions in Trecode.

| Role | Vocabularies Ontologies | Workflows Parameters | Samples Metadata | Genomics Analysis | System | Trecode Users |
|---|---|---|---|---|---|---|
| Admin | R + W | R + W | R + W | R + W | R + W | Admins |
| Manager | R + W | R + W | R + W | R + W | R | bioinformaticians |
| Editor | R | R | R + W | R + W | - | operators, bioinformaticians |
| Viewer | R | R | R | R | - | (wet-lab) researchers |

R = Read, W = Write, - = invisible.

For example, operators need to write permissions for experiment and analysis metadata tables in order to upload new sample metadata or register their performed analyses. Other tables, such as vocabularies, workflow, and workflow compute tables, only require read permissions. System administration entities in Trecode not relevant for data analyses may even be completely invisible to operators. Wet-lab researchers typically use the platform to view sample relations and computational analyses results and are therefore granted read access to those tables. Bioinformaticians responsible for implementing analysis parameter optimizations and bugfixes have write permissions to all genomics data analyses tables. Making changes in the table structures, data model, user/role permission settings, and GUI configuration requires access to the system tables, which is restricted to administrators.

Row level security in Molgenis has recently been fully developed and is being implemented in Trecode for shared tables that contain sensitive data. The user's profile (roles and group memberships) is determinant for obtaining access to a particular row in these tables. Having controlled access in place allows one to open up the platform to a wider audience and encourages (meta) data sharing without sacrificing data privacy and security.

### 3.4. Data Governance

Enforcing data structure and terminology standards is crucial for data reusability and interoperability, as described in the FAIR guiding principles [20]. In addition, adhering to these guidelines opens possibilities to increase the automation of data analysis, making the system more scalable. We structured the data by unambiguously describing entity relationships using a common data model similar to SRA/EGA [13,15]. Entities and attributes in this model are annotated using well-established and maintained ontology terms, which provide computers with the meaning of the data objects. Currently, Trecode has an ontology reference for 479 of the total 524 attributes, and 58 of the 64 entities.

Trecode provides three routes of metadata entry; (i) Operators can interactively insert rows in the database tables, (ii) upload metadata in comma separated or excel files via the web GUI, or (iii) upload metadata programmatically via the REST interface. In all cases, the metadata is checked for consistency and adherence to the implemented data standards, ensuring compatibility with the SRA/EGA [13,15] data model.

Data integrity and reusability of outsourced data storage [22] is enforced through the md5 checksum and use of standardized metadata-encapsulated output formats such as bam, cram, hdf5, and g(vcf) [23–25]. During data analysis, workflows check the sample and file metadata for file integrity and to detect sample-file header inconsistencies. Furthermore, files in Trecode are required to have a Uniform Resource Locator (URL) using an

open standard protocol. This ensures that data in the platform is findable and accessible, regardless if data is stored on a webserver, object store, and/or cloud bucket supporting the transfer of computational analyses from on-premises to cloud compute environments.

All workflows include the tasks responsible for systematically storing analysis results and linking them to the sample and experiment metadata. The goal is to build a track-and-trace system for (bio)materials and their (analysis) products, ensuring data provenance. The origin of analysis result entities is made transparent on the basis of consistent and descriptive labels and titles that include the Trecode identifiers involved and the experimental context. In addition, analysis result file names are automatically and systematically composed by concatenating identifiers from the Trecode entities used and are extended by a unique Trecode analysis id and experimental context. This results not only in human readable file names, but also in sufficient context provided to the user/operator indicating what information can be expected in the file.

Records that are created in sample, NGS experiment, or analyses result-related tables have unique human readable and context traceable identifiers describing the source and type (see methods section).

### 3.5. Data Analysis

Bioinformatic analyses for variant detection in WGS and WXS data is a multistep and multi-software process. Analyses are often performed in a stepwise manner, with the output of one analysis being used as the input for the subsequent step. These analyses workflows are often designed to run automatically without human intervention. Bioinformatic workflows for variant discovery in NGS data typically includes steps that perform: (1) sequence read quality control, (2) read alignment to reference, (3) single-nucleotide polymorphism (SNP) and small insertion or deletion (indel) detection, (4) variant filtering, and (5) variant annotation (Figure 3).

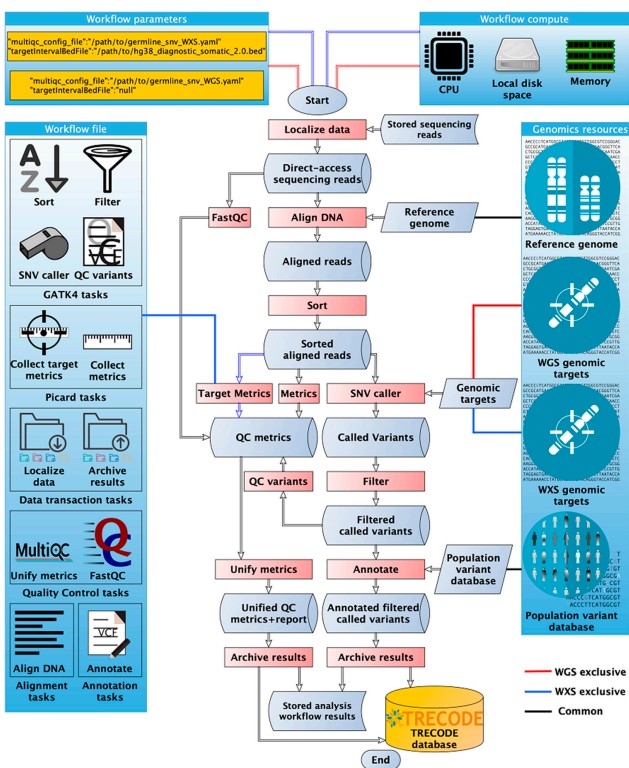

**Figure 3.** Registration and re-use of genomics resources as well as workflow and task scripts in Trecode: (center) Representation of a germline variant detection workflow in which the processing steps are indicated in red; (left) The processing tasks are non-redundantly grouped by software or

function, described and stored in the table 'Workflow_file', and where applicable reused in other workflows; (right) Required reference genome, the genomic targets and population variant databases are described in the table 'Genomics_resources'. The workflow is used to analyze both whole exome and whole genome data; (top left) Workflow parameters are analysis context-specific and, e.g., determine which genomic targets will be analyzed and if target-specific metrics will be collected; (top right) Context-specific computational parameters are stored in the table 'workflow_compute'.

Due to the high similarity in variant discovery between WGS and WXS, the analysis workflows can be shared. However, errors in the sequencing experiment setup and platform biases differ per sequencing context and might require specific QC steps to monitor and detect deviations. In addition, individual steps in a workflow may have specific requirements for genomics resources. The computational resources for running a WXS or WGS workflow might also differ by an order of magnitude due to differences in data volume. Nonetheless, the similarity of WXS and WGS variant discovery is considerable, allowing for a common workflow definition, as long as it has a modular design and is highly configurable. This concept can be extended to the task level to further exploit similarity in omics data analyses by creating generic and configurable task definitions that can be reused across multiple analyses contexts.

### 3.5.1. Integrated Data Analysis Workflows

Currently, the Genome Analysis Toolkit (GATK) best practices [26] sequencing analyses-based variant detection workflows have been implemented in Trecode. These include germline SNV, somatic-SNV, and somatic-CNV. In addition, we have implemented transcriptome analysis for SNVs (GATK4 [27]), quantification, and gene fusion detection (STAR-Fusion [28]). All of the aforementioned workflows generate QC metrics that are merged and uniformly parsed using MultiQC [29]. For performing sequencing provider and platform-specific noise reduction in somatic variant detection workflows, Trecode supports queries for creating, storing, and (re)using panels of normals [21]. The workflows that are currently under active development, and already supported in Trecode, are structural variant detection and DNA methylation profiling. For data visualization and publishing purposes, the Trecode platform includes a workflow that outputs the results of the somatic variant detection analyses in a format that is compatible with the visualization platform cBioPortal [30] and a workflow for automated data submission to EGA [13] using the REST protocol.

The workflows are all designed to be modular. In most cases, the workflow task definitions are imported from so called task containers that are listed as workflow dependencies and registered in the workflowfile table (Figure 3). Task definitions are generic and configured per workflow instance using key value pairs in the tables 'workflowparameters', 'workflowcompute', and 'workflowresources'. For example, WGS and WXS germline analyses refer to the same workflow definition code at workflow instance level. The values for workflow parameter genomic targets (targetIntervalBedfile) and qc config (multiqc_config) determine if the step HSmetrics (hybridization metrics), which is defined in the task container "Picard tasks", is performed and what QC metrics are collected and how they are presented. Likewise, tasks have configurable runtime compute resource requests that are defined per workflow instance and can be optimized per analysis context. For example, the workflow step to identify SNV's in the task container GATK4 is used for both WGS and WXS, but the memory, cpu hours, and tmp diskspace differ between sequencing strategies. In addition to the modulated use of tasks, complete workflows can also be called as a subworkflow. For example, variants identified in an SNV workflow are annotated with VEP [31] using a generic Vep workflow that is called as a sub workflow and controlled by a number of analysis context-specific parameters. This same annotation workflow can be used for both germline SNVs and somatic SNVs as well as SNVs identified in RNA data. This exemplifies that in Trecode the reuse of analysis workflow code is not restricted to task level but can also be used in workflow nesting, which offers high flexibility and low code redundancy in creating new workflows.

### 3.5.2. Analyses Reproducibility by Automation and Tracking System

A fundamental issue with inter-project comparisons and long-term studies is the reproducibility of pipeline results. To achieve this, we have automated the data analyses and implemented a computational analysis tracking system in Trecode (Figure 4).

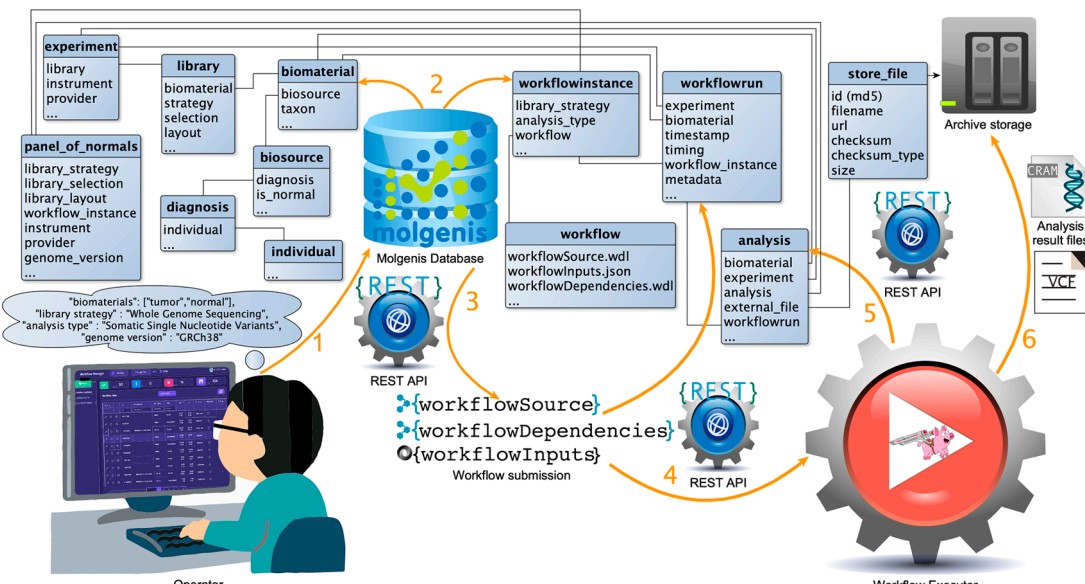

**Figure 4.** Highly automated genome analyses in Trecode. (1) Genome analysis is started by providing the Workflow Manager or Trecode's python command line client with a biomaterial, a library strategy (WGS/WXS/RNA-Seq), an analysis type, and a genome version; (2) The client program queries Trecode using the Molgenis REST interface for run data and a computational analysis workflow being the core workflow 'workflowSource', an input template 'workflowInputs', and generic reusable workflow tasks 'workflowDependencies'; (3) The inputs template is filled in by the command line client using queried experiment metadata and, together with workflow core and dependencies, is submitted via a REST protocol to the Cromwell workflow executer; (4) Workflow execution is registered in Trecode as a 'workflowrun' record, and successful workflow runs will have analyses records attached; (5) Analyses records are created by results archiving workflow tasks and include references to analyses inputs as well as result files ensuring data provenance; (6) All workflows include (shared) tasks for systematic archival of analysis results.

The analyses metadata provides detailed information on how data were created, including which tool versions, parameters, and genomics resources were used during data analyses. In addition, Trecode implements workflow instance versioning across workflow and task definition code as well as logging per workflow instance which software versions, parameters, and genomics resources are used. This provides a complete audit trail of developer and operator activity across all analyses (Figure 4). In some cases, default parameters may need to be overridden, which is a functionality that is provided by Trecode's client interfaces. These overrides, being part of the workflow execution information, are also included in the audit trail. For long term capturing of workflow execution parameters and statistics, we have equipped Trecode with a persistent implementation of the Cromwell workflow executer [10] (see methods section).

### 3.5.3. Workflow Manager

The execution and monitoring of NGS analysis workflows is complex due to the many steps, the execution time of workflows, and interaction between multiple samples. To maintain scalability and assist operators, Trecode provides the operator with a well-arranged interface, named 'Workflow Manager', which communicates via REST API with Molgenis and Cromwell (Figure 5).

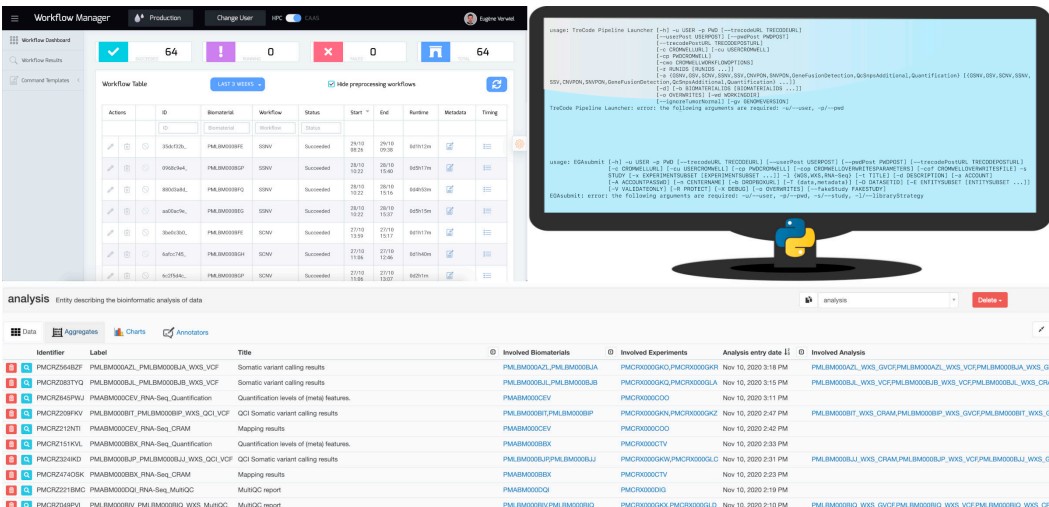

**Figure 5.** Interfaces for operators, bioinformaticians, and researchers. (top left) Workflow manager is a dashboard for operators that provides them with an overview on their daily work. (top right) Trecode client scripts help bioinformaticians in performing sensible analyses; (bottom) Molgenis Web-GUI is an easy to use graphical Trecode querying tool for researchers involved in biobanking and clinical diagnostics.

Within the Workflow Manager, routine analyses can be easily started and monitored for progress. In addition, via this web interface, links to completed analysis files can be retrieved and failed analyses cleaned up. The Workflow Manager is aimed at streamlining routine analysis as an alternative to the more versatile, but less intuitive, Trecode command line client.

## 4. Discussion

By combining the open source initiatives Molgenis [12] and Cromwell [10], we have created Trecode, a (meta) data warehouse and analysis infrastructure that can be applied in both NGS molecular diagnostics as well as research activities. As part of this shared infrastructure, we developed a data model in which diverse genomics experiment designs and analyses and their inter-relationships can be described. The tight integration of sample metadata with computational analyses assists operators in performing sensible analyses and provides an audit trail of parameters used and analysis results generated. Unique to the platform is the flexibility of the metadata model provided by Molgenis [12], together with an agnostic compute backend provided by Cromwell [10] and not seen in other NGS analyses platforms (see Table S2). The flexibility of the data model allows further integration of experiments such as (single cell) transcriptome sequencing and array-based methylation analysis with minimal effort.

In line with current international efforts of standardizing workflow descriptions [11], analysis workflows in Trecode are written using WDL [7] and are executed by the Cromwell workflow executer [10]. When generating workflow code, our emphasis is on reuse, which has resulted in a compact non-redundant and well documented code base that is easy to maintain, extend, and reuse. In order to address the challenge of sharing the bioinformatics tools and produce standardized and completely reproducible analyses [11], container-based software stacks [32,33] are used, resulting in a code base free of local compute infrastructure dependencies. The final aim is making our workflows lab/site agnostic and cloud compute ready. This will additionally provide options to scale-out to the cloud and move workflows to data sources instead of the current approach of moving data to compute workflows.

Metadata are essential to understanding, interpreting, and evaluating genomic assays, including the reuse of analyses results. By committing to FAIR data [20] principles, we aim for complete data reusability. This will further be achieved by metadata extensions increas-

ing the use of defined terms from ontologies such as SNOMED CT (clinical) [34], NCIT (translational research) [35], OBI (biomedical investigations) [36], EFO (experiments) [37], and EDAM (bioinformatics) [38]. Using curated definitions that are logically connected facilitates other researchers to computationally explore and understand our data and finding purposes for re-use.

Collecting rich metadata is a challenge in itself as it often needs to be retrieved from multiple disciplines and sources and includes items that researchers do not deem relevant for their own research questions. We think that a platform such as Trecode that (1) enforces metadata collection at sample registration, (2) facilitates metadata handling through structure and integration, and (3) facilitates programmatic access to metadata shows its added value. Investing in the collection of high-quality metadata is indispensable for driving the use of in-house or public data portals and results in improved data citation and credit in the scientific field.

A number of NGS data management and analysis platforms are available, each with its specific features and capabilities (Table S2). Most platforms support per workflow reproducibility of standard data analyses but lack the data provenance features of Trecode for analyses that involve two or more subsequent workflows. Analyses parameter tuning on standard analyses is well supported by the Trecode, HTS-Flow [39], Closha [40], and Terra [41] platforms. Analyses parameters can be modified, but these changes are only recorded in the Trecode and Terra platforms by their workflow executer. Having a complete audit trail for future reference is a key feature in reproducible bioinformatics. Trecode is the only platform that offers the flexibility of allowing any parameter adjustment at the moment of workflows execution using a parameter override function in the command line client and workflow manager GUI while still providing an audit trail. Moreover, all workflow resources and parameters are stored and annotated in Trecode, which makes it easier to find their meaning and encourages re-use across different workflows. Operators and lab staff welcome the integrated experiment QC and computational error reports provided exclusively by Trecode. However, analysis result visualization such as that provided in OTP [42] and QuickNGS [43] is limited in Trecode. From the point of view of good software design, we separated the presentation layers by delegating result visualization to dedicated platforms such as cBioportal [30] and R2 [44] in pediatric oncology.

Trecode and OTP are the only platforms with a (meta)data model that shows similarity to SRA/EGA. Data model similarity is required for automated data submission and ingestion without the need for manual completion and formatting of the data and metadata. With some effort, such a data model can be built in Terra, but functionality via a REST API and web-based navigation is lacking. Flexibility in adapting and extending the data model is superior in Trecode because it is built on Molgenis [12], which provides a framework to generate interoperable platforms based on a custom data model.

The Trecode platform demonstrates how FAIR data principles can be implemented in the context of NGS sequencing, harmonize (meta)data capture and representation, and facilitate large-scale data (re)use. The platform can be used in a combined clinical and research setting to maximize the translation of knowledge between research and healthcare, while addressing the requirements of all stakeholders, including patients/participants whilst addressing ethical, legal, and social aspects. The handling, analysis, and annotation of patient data soon finds itself at odds with the will of the patient and the increasing privacy legislation that also differs between countries. To prevent unauthorized use of data, we have found it useful to register the patient's informed consent status in the platform, which will be added in the next release. In addition, it is essential to use non-traceable identifiers in systems such as Trecode to prevent a direct link between sensitive genome information and a person. Several similar endeavors are being undertaken to promote NGS data and knowledge transfer. Recently, guidelines have been drafted by multidisciplinary delegates from academic medical and research centers to facilitate large-scale (re)use of all human genomic data in the Netherlands [45]. By announcing Trecode and making the code and software stack available we aim to narrow the gap between the

proposed far-reaching recommendations and its practical implementation in a clinical and research setting. With the platform that we present and share here, we want to motivate research centers to make experimental data and computational analysis pipelines FAIR at the source. Starting with FAIR data implementation can be a big hurdle that is now easier to overcome. The Ansible [46] code for an automated platform rollout is available at https://bitbucket.org/princessmaximacenter/trecode (accessed on 24 November 2022) and only requires a medium-sized (virtual) machine on which the whole platform can be deployed. Feedback and contributions to this repository are greatly appreciated and will feed into subsequent releases of the Trecode platform.

**Supplementary Materials:** The following supporting information can be downloaded at: https://www.mdpi.com/article/10.3390/biomedinformatics3010001/s1, Figure S1: Data model for enrolling patients to studies.; Figure S2: Description of DNA-methylation experiments in Trecode data model.; Figure S3a: Automated data submission to EGA.; Figure S3b: Data model entities related to EGA submission.; Table S1: Trecode metadata type by two letter code identifiers; Table S2: Feature comparison of currently available genomics (meta) data management and processing platforms; Supplementary Figure Legends.

**Author Contributions:** Conceptualization, H.H.K., J.Y.H.-K., and P.K.; methodology, H.H.K., J.Y.H.-K., and E.v.d.G.; software, H.H.K., J.Y.H.-K., E.v.d.G., C.v.R., S.B., A.J., J.B.-H., D.v.d.L., and E.T.V.; validation, H.H.K., J.Y.H.-K., E.v.d.G., and C.v.R.; formal analysis, H.H.K., J.Y.H.-K., E.v.d.G., S.B., A.J., J.B.-H., S.d.V., D.v.d.L., and E.T.V.; investigation, H.H.K., J.Y.H.-K., E.v.d.G., C.v.R., S.B., A.J., D.v.d.L., and E.T.V.; resources, B.B.T. and P.K.; data curation, H.H.K. and J.Y.H.-K.; writing—original draft preparation, H.H.K.; writing—review and editing, J.Y.H.-K. and P.K.; visualization, H.H.K.; supervision, B.B.T. and P.K.; project administration, H.H.K., J.Y.H.-K., B.B.T., and P.K.; funding acquisition, P.K. All authors have read and agreed to the published version of the manuscript.

**Funding:** This research was funded by Stichting Kinderen Kankervrij and Adessium Foundation.

**Institutional Review Board Statement:** Not applicable.

**Informed Consent Statement:** Not applicable.

**Data Availability Statement:** The code and automated deployment recipe of the Trecode public release is openly available at https://bitbucket.org/princessmaximacenter/trecode (accessed on 24 November 2022).

**Acknowledgments:** We gratefully acknowledge useful feedback and discussion with lab members and collaborators within the FAIR genomes project.

**Conflicts of Interest:** The authors declare no conflict of interest.

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
