# Peer review of "Trecode: A FAIR Eco-System for the Analysis and Archiving of Omics Data in a Combined Diagnostic and Research Setting"

_biomedinformatics, doi:10.3390/biomedinformatics3010001_

Round 1
Reviewer 1 Report
Figure need to be redone. Figure 1 is too small to be read (even with magnifying glass). Tis should be one entire page.
Same for Figure 2.
Figure 3 could be enlarged in space allows.
What is the overall purpose of the paper? The abstract sayd (last sentence) that Trecode etc. are publicly available. If the purpose of the paper is to try to have other groups get Trecode and use it, the conclusions should include a paragraph or two emphasizing that point, which I believe is important. Extra information about whom to contact, what is required to implement it, etc. would be very useful.
I suggest that at sme point the authors address how they handle privacy concerns about the "Individuals" about whom data is collected and stored.
Im paragraph 2.3 "Identifiers" require either familiarity with the Encode project or access to the reference. This paragraph could be reformatted to clarify the patterns.
Paragraph 3.1 line 142-143 mentions "community-developed standard formats." A reference would be useful. Same about well-established ontologies.
3.4.1 line 375. While GATK is "well-known", I suggest such acronyms bedefined first time used. Applies to other situations.
The above suggestions are intended the breadth of relevance for the paper beyond NGS data handling. The concept of brings data analysis to the data rather than the opposite is worth sharing with broader bioinformatics and other scientific communities.
Author Response
We thank reviewer1 for the feedback on the manuscript and have taken the suggestions to improve the manuscript:
We agree with Reviewer1 that the overall aim of the paper, promoting FAIR data at the source, can be better served.
In the abstract and in the discussion, we now emphasized that by providing an automatic roll-out of Trecode, we want to support the research labs in making their data and analyses FAIR.
As mentioned by reviewer1, patient privacy is compromised as soon as anonymous genomic data is linked to patient metadata. In the discussion section of the manuscript, we emphasized that in platforms such as Trecode only identifiers that cannot be traced back to the patient should be used. In addition, including the patient's will (consent) in the data model has proven to be of added value to prevent unauthorized use of data and will be part of the data model in the next release.
We agree with reviewer 1 that the readability of paragraph 2.3 (“Identifiers”) can be improved and have therefore rewritten it
At the request of reviewer 1, we referenced sources of "community-developed standard formats" and "well-established ontologies" (paragraph 3.1)
We checked the manuscript for unexplained acronyms and declared them.
We have also included higher resolution figures and will confer with the editorial office for an optimal layout of Fig. 1, 2 and 3.
Reviewer 2 Report
My only comment/query is minor: In Table S1 should the spelling "Biosourse" should be corrected to read "Biosource"?
Author Response
We thank reviewer 2 for reviewing the manuscript and have corrected the spelling error in tableS1.
Reviewer 3 Report
Authors developed a very useful resource that has huge potential to manage and integrate the clinical, experimental, and computational workflow. Such workflow can be utilized for maximum use of data in a complete, integrated, and shared fashion. It is expected that authors will continuously improve the integration workflow of the tool in the future based on user experience.
Author Response
We thank reviewer 3 for the positive feedback and will indeed continue to improve the Trecode platform based on user experiences.